# Introducing the Role of Genotoxicity in Neurodegenerative Diseases and Neuropsychiatric Disorders

**DOI:** 10.3390/ijms25137221

**Published:** 2024-06-29

**Authors:** Glen E. Kisby, David M. Wilson, Peter S. Spencer

**Affiliations:** 1Department of Biomedical Sciences, College of Osteopathic Medicine of Pacific Northwest, Western University of Health Sciences, Lebanon, OR 97355, USA; 2Biomedical Research Institute, BIOMED, Hasselt University, 3500 Hasselt, Belgium; david.wilson@uhasslet.be; 3Department of Neurology, School of Medicine, Oregon Institute of Occupational Health Sciences, Oregon Health & Sciences University (OHSU), Portland, OR 97239, USA

**Keywords:** genomic instability, genomic stress, DNA damage response (DDR), DNA repair, environment

## Abstract

Decades of research have identified genetic and environmental factors involved in age-related neurodegenerative diseases and, to a lesser extent, neuropsychiatric disorders. Genomic instability, i.e., the loss of genome integrity, is a common feature among both neurodegenerative (mayo-trophic lateral sclerosis, Parkinson’s disease, Alzheimer’s disease) and psychiatric (schizophrenia, autism, bipolar depression) disorders. Genomic instability is associated with the accumulation of persistent DNA damage and the activation of DNA damage response (DDR) pathways, as well as pathologic neuronal cell loss or senescence. Typically, DDR signaling ensures that genomic and proteomic homeostasis are maintained in both dividing cells, including neural progenitors, and post-mitotic neurons. However, dysregulation of these protective responses, in part due to aging or environmental insults, contributes to the progressive development of neurodegenerative and/or psychiatric disorders. In this Special Issue, we introduce and highlight the overlap between neurodegenerative diseases and neuropsychiatric disorders, as well as the emerging clinical, genomic, and molecular evidence for the contributions of DNA damage and aberrant DNA repair. Our goal is to illuminate the importance of this subject to uncover possible treatment and prevention strategies for relevant devastating brain diseases.

## 1. Introduction

The collection of papers in this Special Issue focuses on the molecular mechanisms that give rise to neural cellular dysfunction and network deterioration in several neurodegenerative and neuropsychiatric disorders. Understanding the underlying disease mechanisms is complicated by the dearth of definitive etiological information on these diverse conditions, i.e., the initial triggers that set in motion the cascade of molecular steps that result in brain dysfunction and clinical illness. The triggers may be of endogenous origin, such as an inherited mutant gene, a persistent genomic alteration, or a metabolic impairment, resulting in elevated levels of potentially neurotoxic molecules and/or neuroactive products. The triggers may also be of exogenous origin, such as exposure to an infectious agent that initiates an autoimmune response within the brain or to a chemical with neurotoxic potential that acts alone or on a genetic background that increases susceptibility to an environmental factor to promote disease. Additional challenges in understanding disease etiology include determining (i) when the trigger occurred, i.e., in utero, infancy, childhood, adolescence, or adulthood, and (ii) to what extent the neural cellular changes that accompany biological aging contribute to disease risk and development. Notably, features characteristic of one disease class (neurodegenerative or neuropsychiatric) may appear to some extent in the other with overlapping molecular mechanisms, whether at the gene level [1], post-translational modifications, signaling events, or in relation to the microbiota–gut–brain axis [1,2].

Neuropsychiatric disorders often have developmental origins with phenotypic expression early in life. Conversely, the timing and triggers of neurodegenerative disorders are often difficult to pinpoint since they are mostly expressed clinically in the second half of life after a long latent period. Nonetheless, a neurodevelopmental event is thought to contribute to the development of both psychiatric disorders, such as schizophrenia, bipolar disorder, and obsessive-compulsive disorder, and neurodegenerative diseases, such as Parkinson’s disease (PD) and Huntington’s disease (HD) [3]. Neuropsychiatric and neurodegenerative disorders may result from individual or a combination of genetic variants, single or multiple environmental exposures, or a specific collection of genetic and environmental factors acting in concert [4,5] (see Figure 1). Western Pacific Amyotrophic Lateral Sclerosis-Parkinsonism-Dementia Complex (ALS-PDC), a prototypical progressive neurodegenerative disorder with neuropsychiatric features [6], has a predominantly environmental etiology but with a latency between exposure and clinical presentation that can span decades (discussed in more detail below). The above complexities in disease etiology have provided the stimulus for exploring the role of genotoxicity in neurodegenerative disease and neuropsychiatric disorders in this Special Issue. The present review does not seek to provide a comprehensive review of the subject but rather to highlight the role of genomic integrity in these disorders and to provide a foundation for encouraging further investigation in this field.

### 1.1. Neurodegenerative and Neuropsychiatric Disorders Overlap

There is clinical, epidemiological, and biological evidence of an overlap in the expression of several neurodegenerative and neuropsychiatric disorders, a subject reviewed in detail by Seritan [7]. For example, paranoid delusions affect many with Alzheimer’s disease (AD). Visual hallucinations and delusions (including Capgras syndrome) occur in dementia with Lewy bodies. While minor hallucinations may predate the onset of motor signs in PD, psychotic features can occur later in association with, or exacerbated by, drug treatment. The prodrome (i.e., an early sign or symptom that indicates the onset of disease) of HD is often featured by apathy, depression, irritability, and anxiety, while psychosis, aggression, and suicidality may affect many HD patients over their lifetime. Behavioral manifestations of frontotemporal dementia include apathy, loss of empathy, disinhibition, hyperorality, and compulsive behaviors, along with anxiety, depression, and executive dysfunction. Paranoid ideation, delusions, or mainly visual hallucinations affect many patients with frontotemporal lobar dementia; this is yet another example of coincident manifestations in neurodegenerative and neuropsychiatric brain disorders.

Perhaps the best example of such overlapping expression is the association between amyotrophic lateral sclerosis (ALS) and psychiatric illness, particularly schizophrenia [8]. For example, Westphal [9] reported that schizophrenia, paranoia, and manic-depressive states are associated with ALS, and Wechsler and Davison [10] noted that these mental symptoms arise due to cortical degenerative changes. Moreover, Turner and colleagues [11] found that schizophrenia may represent a risk factor for ALS (OR 5.0), and Howland [12] reported several cases in which schizophrenia occurred in ALS patients. A register-based nationwide study in Sweden found a higher occurrence of schizophrenia up to 1–5 years before and 2–5 years after ALS diagnosis [13]. Misdiagnosis aside, the coexistence of ALS and schizophrenia has been interpreted as having a shared polygenic basis [8], with an estimated genetic correlation of ~14%. GWAS studies also suggest a genetic correlation between the two conditions [14]. Additionally, other neuropsychiatric conditions, such as obsessive-compulsive disorder, autism, and alcoholism, occur more frequently in first- or second-degree relatives of ALS patients with and without *C9Orf72* expanded repeats [15,16], the most common genetic cause of ALS and related disorders of the ALS/frontotemporal lobar degeneration spectrum [17]. Disturbances in motor neuron function have been demonstrated in schizophrenia [18,19,20], further suggesting an overlapping pathophysiology.

Taken in concert, the foregoing examples justify the coincident analysis of the etiology of neurodegenerative and neuropsychiatric disorders, whether of genetic, environmental, or mixed origin. We focus henceforth on the role of genomic instability in these potentially overlapping neuropathological outcomes.

### 1.2. DNA Damage, Genomic Instability, and DNA Repair

Continuous, unremitting damage to one’s genetic material is unavoidable, arising via reactions with endogenous chemical species or via direct or indirect interactions with external agents [21]. The best-known endogenous genotoxins encompass a collection of reactive oxygen and nitrogen species (RONS), predominantly produced as byproducts of mitochondrial oxidative phosphorylation, as well as aldehydes, alkylators, and mobile DNA elements. Another prominent endogenous molecule with genotoxic potential is formaldehyde, which normally regulates one-carbon metabolism [22]. Environmental DNA-damaging agents span sunlight, ionizing radiation, many naturally occurring and manmade chemical compounds (including formaldehyde), and RNA/DNA viruses, to name a few. Via these endogenous and exogenous mechanisms, the genome can be modified in ways that alter base, sugar, or phosphodiester bond composition and integrity. Depending on the nature of the DNA damage, the accuracy or the operation of DNA transactions—namely transcription or replication—can be adversely affected. Resulting mutagenesis, genomic instability, or transcriptional or replicative stress can promote cellular transformation, senescence, or death, outcomes that underpin pathologies such as cancer, degenerative disease, or accelerated aging [23].

Given the potentially severe adverse effects of unrepaired DNA damage, organisms have evolved eloquent protective systems, collectively known as DNA-damage response (DDR), that recognize and resolve the many types of genomic stress, preserving genome integrity and cellular health. The main nuclear DNA repair mechanisms [24], many of which are coupled to signaling pathways that regulate cell cycle checkpoints to permit efficient response time in replicating cells, include the following: direct reversal (DR), which encompasses a collection of proteins (namely, the *O^6^*-methylguanine-DNA methyltransferase (MGMT) and the alkylated DNA repair protein B (AlkB) homologs) that directly resolve primarily base modifications without the need for DNA degradation and reconstruction [25]; mismatch repair (MMR), a pathway that copes with DNA replication errors, e.g., mismatched bases or small insertion/deletion loops [26]; ribonucleotide excision repair (RER), a process that removes inadvertently inserted ribonucleotides from genomic DNA [27]; base excision repair (BER), a system that copes with many simple spontaneous or oxidative base or sugar lesions [28]; nucleotide excision repair (NER), a mechanism composed of general genome and transcription-coupled sub-pathways that resolve bulky, helix-distorting base adducts [29]; and recombinational repair, a term that encompasses both homology-directed homologous recombination (HHR) and non-homologous end-joining (NHEJ) pathways [30]. By and large, the above repair systems entail coordinated steps of recognition, processing, resolution, and restoration to preserve the original state of the genome. Not surprisingly, inherited or sporadic defects in DDR components result in increased disease manifestation, most notably cancer predisposition and neurological disease, as well as accelerated aging phenotypes.

Depending on various cellular characteristics, e.g., replicative status, metabolic activity, etc., the different pathways take on varying levels of importance in maintaining cellular homeostasis. In particular, while the different DNA repair mechanisms introduced above are broadly operational in cycling cells, e.g., neural progenitors, upon terminal differentiation and the establishment of a non-replicating status, systems like BER (and its related pathway of single-strand break repair; SSBR), NER, and NHEJ are thought to take on a greater role [31]. This increased responsibility largely stems from the obvious mechanistic links between MMR, RER, and HHR with the DNA replication machinery and its activities. Moreover, in the specific case of mature neurons, which possess a high energy demand and, therefore, carry out a high level of RONS-generating oxidative phosphorylation, pathways such as BER and NER, i.e., the primary systems for resolving oxidative DNA damage, take on an even greater importance. In addition, studies have found that neuronal MGMT levels decline markedly after terminal mitosis, making the non-cycling cell more vulnerable to endogenous and exogenous agents that generate alkylative DNA damage [32,33,34]. Thus, depending on the timing of DNA repair complications or a genotoxin exposure, i.e., during the early stages of development (active neurogenesis) or later in life in mature adults, the molecular, cellular, and pathological neural system outcomes could look very different. We expound upon this often-overlooked element in the discussions that follow.

### 1.3. Genomic Instability in Neurodegenerative Diseases

Genomic instability can be defined as an increased likelihood of experiencing genomic alterations arising from the accumulation of DNA damage that results from elevated genotoxic exposure or a defect in the resolution of DNA damage [35]. While genomic instability, i.e., the resulting genomic alterations (e.g., point mutations, insertions/deletions, and chromosome aberrations that arise), is an established hallmark of cancer and aging, its relevance to the underlying pathogenesis of progressive neurodegenerative disease is not completely understood [36]. The neurodegeneration that is observed in inherited DNA-repair disorders (e.g., Ataxia Telangtasia (A-T), xeroderma pigmentosum (XP), Cockayne syndrome (CS)) has provided evidence for the importance of maintaining genomic stability during brain development and following maturation [31,37,38]. A-T is an autosomal recessive disease characterized by progressive neurodegeneration, as well as other non-neurological symptoms (endocrine and immune dysfunction), stemming from a mutant form of a protein kinase (ATM) that is an established regulator of the DDR and an important sensor of oxidative stress [39,40,41,42,43,44]. ATM is a serine/threonine protein kinase that is recruited and activated by DNA double-strand breaks (DSBs), topoisomerase cleavage complexes, DNA-RNA R-loops, and, in some cases, DNA SSBs. The broader importance of persistent DNA strand breaks, particularly SSBs, in neurological disease, which would arise via impaired neurogenesis, post-mitotic neuronal cell loss, or both, is supported by a collection of disorders with inherited mutations in core DNA repair factors, such as ataxia with oculomotor apraxia (AOA) types 1 and 2 and spinocerebellar atrophy with axonal neuropathy type 1 (SCAN1), amongst others [43,44]. About 20% of XP patients develop progressive degeneration of cortical, basal ganglia, and cerebellar neurons, as well as spinal atrophy, cochlear degeneration, and axonal neuropathy [45,46]. The clinical manifestations of CS include substantial growth defects, neuronal loss, calcification, mental retardation, and postnatal microcephaly [47]. Cells from both XP and CS patients are defective in NER of actively transcribed genes (*TC-NER*), while XP cells are also defective in global genome NER (GG-NER), both processes that likely take on vital roles in neural cells. Defects in other DNA repair pathways (i.e., BER) might also contribute to the mitochondrial dysfunction observed in CS patients [48]. Thus, analysis of the aforementioned DNA-repair syndromes demonstrates that pathways that recognize DNA damage (DDR) and directly repair the DNA damage are crucial for maintaining the integrity of the neural genome to prevent neuronal loss or dysfunction not only during development but also in the mature brain.

Several studies suggest that the loss of genomic integrity is an important trigger of the progressive neuronal dysfunction in age-related neurodegenerative diseases, such as Alzheimer’s disease (AD), PD, and ALS [49,50,51,52]. Transcriptome and epigenome data from single-cell studies of AD brain tissue have revealed that DNA damage and DNA repair increase substantially as the disease progresses [49,53]). In line with prior observations [54], Dileep and colleagues [49] demonstrated that persistent DSBs are an early pathological hallmark of AD based on their detection of gene fusions in both the AD brain and an AD mouse model. Using a cut-and-run method to detect the genome-wide distribution of DSBs in AD (n = 3) and age-matched control (n = 3) brains, the former was observed to contain 18 times more DSBs, and this DNA damage also correlated with AD-associated SNPs, increased chromatin accessibility and gene expression [55]. The identification of persistent neuronal DSBs and the associated activation of DNA repair are indicators that genomic instability likely contributes to the progression of AD, along with the observed oxidative stress and macromolecular (DNA) damage [54,56,57]. Collectively, these recent studies demonstrate a loss of genome integrity in AD brain cells that is due to DNA strand breaks and imbalanced DNA repair mechanisms that might cause epigenomic dysregulation in brain cells. Thus, these cellular events may explain the loss of cell identity in AD and the ensuing cellular senescence and loss of neuronal function.

There is growing evidence that DNA damage and DDR also play an important role in the underlying pathogenesis of PD [50,58]. Misfolded α-synuclein (a pathological hallmark of PD) induces mitochondrial and genomic DNA damage in microglia, subsequently activating the downstream c-Gas STING pathway, a key mediator of inflammation in the settings of infection, cellular stress, and tissue damage [59]. Both genotoxic damage and microglial STING activation were reproduced in mice after intrastriatal injection of misfolded α-synuclein, features observed in PD brain tissue, as well [60]. Thus, microglial-induced genomic DNA damage appears to be an important mechanism for triggering the neuroinflammation and neurodegeneration observed in PD. Moreover, oxidative DNA damage (i.e., 8-oxodG) accumulates in both nuclear and mitochondrial DNA (mtDNA) of nigral dopaminergic neurons in PD [50]. A significantly higher accumulation of 8-oxodG and the mutant α-synuclein protein S42Y was detected in the midbrain of PD subjects (n = 8) compared to controls (n = 9) [61]. The preliminary studies by Basu and colleagues [61] suggest that oxidative DNA damage is likely responsible for the accumulation of S42Y following the misincorporation of adenine opposite 8-oxodG during transcription (i.e., transcriptional mutagenesis, TM). The S42Y protein was also shown to be more toxic to murine cortical cultures and accelerated the aggregation of wild-type α-synuclein. Further studies are needed to determine the link between synuclein proteins and TM in PD. Moreover, DNA damage (especially persistent) can induce transcriptional errors that result in altered protein functions [62], increasing their misfolding and aggregation [63]. Thus, genotoxic stress induced either by endogenous (e.g., oxidative DNA damage) or environmental exposures (e.g., genotoxins) can lead to transcriptional errors (i.e., TM) and subsequent proteotoxic stress [64,65,66], a characteristic pathogenic feature of most progressive neurodegenerative diseases.

Emerging studies of ALS patients indicate that progressive motor neuron degeneration is associated with the accumulation of DNA damage and a deficiency in DNA repair [67]. These early events appeared to induce persistent transcriptional changes after ALS patient-derived stem cells were differentiated into neurons [68]. In 5–10% of patients with ALS, mutations occur in the genes coding for TAR DNA-binding protein 43 (TDP-43) and DNA/RNA-binding protein fused-in sarcoma (FUS) [69,70,71], and both corresponding proteins interact with the transcription-coupled nucleotide excision repair machinery [72]. The high level of transcription in neurons generates the formation of R-loops (naturally occurring RNA/DNA hybrids), which, if they persist, can lead to single-strand breaks (SSBs) and DSBs [73,74]. Recent studies show that R-loops are generated in neurons and cells derived from ALS patients transfected with mutated TDP-43 (A382T) [75]. Thus, TDP-43 pathology in ALS is associated with R-loop-mediated DNA damage. Detection of FUS-induced mtDNA damage and mtDNA repair deficiency in patient-derived induced pluripotent cells also suggests that genotoxic stress plays an important role in the pathogenesis of ALS [76].

### 1.4. Role of Environmental Factors in Neurodegenerative Disease

Environmental factors are increasingly implicated in neurodegenerative diseases, as highlighted in part by recent interest in the role of the exposome [77,78], which spans air pollution, pesticides, metals [79], and hydrazinic compounds [80], among others. Maintaining genome stability involves coordination between different subcellular compartments that provide cells with mechanisms for sensing DNA damage (DDR) and signaling DNA repair systems that safeguard against environmental and endogenous genotoxic stress. Genotoxic stress resulting from endogenous DNA damage is well established in neurodegenerative diseases [55,81,82,83], but the contribution of genotoxic stress following human exposure to environmental chemicals is becoming more recognized as an equally important contributor to the genomic instability observed in age-related neurodegenerative diseases and neuropsychiatric disorders [84,85]. While gene–exposome interaction is often hypothesized in the etiology of neurodegenerative disease, either genetic or, as shown in the following example, the exposome, may have a dominant or, possibly, an exclusive etiological role.

### 1.5. Western Pacific ALS/PDC

Perhaps the best studied human neurodegenerative disorder—although not the best known or acknowledged—is Western Pacific ALS/PDC, a polyproteinopathy of varying phenotype (ALS, Parkinsonism-dementia (P-D), dementia (GD), and sub-clinical neurofibrillary degeneration comprising hyperphosphorylated tau protein). This single disease formerly occurred in high incidence in three genetically distinct populations in the Western Pacific region, including among (i) Chamorros and other Guamanians, (ii) Japanese residents of the Ki Peninsula of Honshu island, and (iii) Auyu and Jaqai linguistic groups living in the southwest lowlands of the island of New Guinea [86]. Extensive epidemiologic and observational studies demonstrated this progressive neurodegenerative disease was acquired early in life but not expressed clinically until years or decades later, with the molecular and cellular events that occurred during the intervening “silent” period being of great importance but virtually unexplored [87]. Post-mortem studies of ALS/PDC brains revealed evidence of nitrative and oxidative stress [88,89], polyproteinopathy (tau, α-synuclein, TDP-43 and sparse *β*-amyloid) [90,91], disturbance of protein homeostasis pathways (ubiquitin–proteasome system and the autophagy–lysosome pathway) [92], and activation of the unfolded protein response [93].

During the second half of the 20th century, ALS/PDC incidence declined in all three affected populations, but it was the absence of a culpable genetic locus [94], the increasing age of clinical onset, and the eventual disappearance of the disease from Guam that confirmed its primary environmental etiology [94]. The decline of ALS/PDC on Guam can be traced to the post-World War II Westernization of cultural practices, specifically to the decline and eventual stoppage of the traditional Chamorro use of, and WWII reliance on, the poisonous seed of the cycad plant for food [86]. Daily use of incompletely detoxified cycad flour in foods exposed consumers to methylazoxymethanol (MAM)—the aglycone of the principal cycad toxin (cycasin)—a potent genotoxin with developmental neurotoxic and carcinogenic potential [34]. Similarly, replacement of the traditional use of cycad seed as an oral tonic/medicine coincided with the late-20th century decline of high-incidence neurodegenerative disease in the Kii-Japan focus of ALS/PDC [95,96]. In addition to the diverse clinical phenotypes of ALS-PDC, in which younger subjects mostly presented with motor neuron disease, P-D appeared largely in middle-aged and GD in older subjects; sometimes, all three phenotypes could be found in individual families on Guam. Variation of three single nucleotide polymorphisms in the *MAPT* (tau) gene correlated with the risk for ALS, P-D, and dementia GD phenotypes on Guam [97]. Guamanian and Kii-Japanese ALS/PDC patients sometimes had a stationary retinal pigmentary epitheliopathy, along with evidence of developmental disruption of the cerebellum, changes that were traced to cerebellar and retinal dysplasia arising during pregnancy and reproduced experimentally in various mammalian species by post-natal treatment with MAM or cycasin [98]. While developmental exposure to the cycad-derived genotoxin was apparent in some patients, others (post-WWII Filipino immigrants to Guam) developed ALS/PDC after first exposure to the Chamorro lifestyle as young adults [99].

Human and experimental evidence indicates a continuum between MAM exposure, brain DNA damage, TM, developmental brain perturbations, and the subsequent appearance of a range of progressive neurodegenerative features [86]. Experimental studies have shed light on what appears to be initial molecular events in the pathogenesis of ALS/PDC. Young adult laboratory animals treated with MAM develop DNA damage (i.e., *O*^6^-methylguanine (*O*^6^MG), *N7*-methylguanine (*N7*MG) adducts) in the liver, kidney [100], and brain, where the active metabolites (i.e., methyldiazonium ion and formaldehyde) induce disease outcomes that depend largely on the replicative status of the affected cell [33]. In cycling cells, unrepaired DNA damage leads to mutation and uncontrolled mitosis, likely promoting carcinogenesis, whereas in postmitotic neurons, cells with excessive damage attempt to re-enter the cell cycle but undergo apoptosis or nonapoptotic cell death, promoting neurodegeneration [80]. In animal studies, once a threshold level of MAM-induced DNA adducts had been reached, there was a detectable transcriptional activation of the DDR pathway [101]. Moreover, MAM-treated young adult mice repaired the genotoxin-induced DNA damage more efficiently in the liver than the brain [33], presumably because post-mitotic neurons are deficient in MGMT, the major repair protein for *O*^6^MG [32,102]. In quiescent neural cells, MAM and related genotoxins induce TM in the absence of detectable DNA mutations [63,66]. While TM can alter gene expression programs, lead to the production of mutant proteins, and alter protein function [62], whether this effect explains the development of brain polyproteinopathy in ALS/PDC has yet to be demonstrated.

### 1.6. Role of Formaldehyde (FA) in Neurodegenerative Disease

Another neurotoxin in the cycad seed, i.e., the free amino acid and cyanotoxin *β*-*N*-methylamino-L-alanine (L-BMAA), also produces a motor system disease in laboratory animals, including non-human primates [103]. Notably, both MAM and L-BMAA are metabolized to formaldehyde (FA), which has both neurotoxic and carcinogenic potential. Exposure to FA by inhalation has been reported to impair memory and cognitive function in humans, to induce deficits in learning and memory, neuronal damage and oxidative stress in the cerebellum of experimental animals, and to induce misfolded neuronal tau and related proteins in vitro [104]. Occupational exposure to FA has also been associated with ALS (but not in all studies), and the chemical impairs olfactory function as well, a symptom noted early in the development of ALS, PD, and AD [104] and Guam ALS/PDC [105]. FA exerts its harmful effects by both damaging DNA and inhibiting DNA repair of O^6^MG, leading to genomic instability [106] and the production of abnormal and potentially misfolded proteins. Moreover, FA-responsive miRNAs predicted to modulate MAM-associated genes in the brains of MGMT-deficient mice include miR-17-5p and miR-18d, which regulate genes involved in tumor suppression, DNA repair, *β*-amyloid deposition, and neurotransmission [107]. These findings bring together cycad-associated ALS-PDC with colon, liver, and prostate cancer; they also add to evidence linking changes in microRNA status both to ALS, AD, and parkinsonism and to cancer initiation and progression. As discussed below, MAM induces a widely used animal model of schizophrenia.

Notwithstanding FA as an environmental contaminant with toxic potential, the compound also serves as an indispensable and, thus, normal physiological metabolite in the healthy brain, where it is proposed to regulate learning and memory via the *N*-methyl-D-aspartate receptor [108]. As a member of the one-carbon cycle, endogenous FA plays a significant role in nucleotide biosynthesis [109] but can also be produced at levels sufficient to pose a significant threat to genomic stability [109] and DNA repair [110]. Endogenous FA can impede transcription, with negative physiological consequences [111] through epigenetic alterations [112], including cancer growth promotion and neuronal, hippocampal, and endothelial damage [113]. Notably, impaired memory has been described in mice with elevated endogenous FA, induced by knock-out of the gene coding for aldehyde dehydrogenase-2, a key mitochondrial enzyme for the effective metabolism of alcohol and acetaldehyde [108]. The balance between genotoxin and benign metabolite is presumed to depend on concentration, localization, pH and redox state [114], features that are definitely or potentially altered during disease progression.

Aging leads to brain accumulation of FA due to defects in its metabolism, and excessive FA directly impairs memory by inhibiting the NMDA receptor [115]. Importantly, endogenous FA levels are increased to some extent in Mild Cognitive Dementia (MCI) and to a greater degree in AD that follows MCI [116]. AD-related *β*-amyloid is proposed to accelerate FA accumulation by inactivating alcohol dehydrogenase-5; in turn, FA promotes Aβ oligomerization, fibrillation, and tau hyperphosphorylation [115]. Indeed, repeated intracerebroventricular injection of FA induces AD-like pathological markers and cognitive impairment in young rhesus monkeys independent of genetic predispositions [117]. Furthermore, the brains of macaques fed methanol, which is metabolized to FA, showed an increase in tau phosphorylated aggregates and *β*-amyloid plaques in four brain regions postmortem, namely the frontal lobe, parietal lobe, temporal lobe, and hippocampus [118]. In sum, the ability of FA to induce genomic instability may be of critical relevance to understanding the genesis of neurodegenerative disease.

### 1.7. Other Environmental Factors

Other environmental factors potentially associated with the induction of genomic instability relevant to neurodegenerative disease include pesticides [119], heavy metals [120], and air pollutants, mainly ozone and nitrogen dioxide [121]. In the case of air pollution (which contains FA), exposure to fine particulate matter 2.5 (PM_2.5_) is proposed to promote organ DNA damage, induce inflammation and oxidative stress in the brain, affect the deposition of *β*-amyloid, promote tau phosphorylation, and serve as a risk factor for AD, especially in subjects with *APOE* ε4 alleles [122,123,124]. Urinary biomarkers of oxidative stress resulting from DNA damage are also found in pesticide applicators and farm workers exposed to organophosphorus (OP) compounds (mostly the anticholinesterase azinphosmethyl), with the amount of DNA damage correlating with the extent of pesticide exposure [125,126]. Most studies have also found positive associations between occupational exposure to complex pesticide mixtures and the presence of chromosomal aberrations, sister-chromatid exchanges, and micronuclei, but several studies failed to detect cytogenetic damage [127]. The long-term brain health consequences of such genomic changes are presently unknown, and they represent a key gap in our knowledge concerning the role of environmental agents in neurological disorders. Nevertheless, people working in agriculture, which is associated with various chemical exposures, reportedly have high rates of brain cancer and PD [128,129], substantially greater odds of developing dementia [130], and increased rates of anxiety, depression, and suicide [131,132,133], thereby emphasizing the importance of mitigating exposure risk.

### 1.8. Role of Neuropathological Proteins in DNA Repair

DNA damage and DNA repair have been shown to be influenced by pathological forms of proteins (TDP-43, FUS, C9Orf72, α-synuclein, and tau) that accumulate in a number of age-related neurodegenerative diseases, including ALS [134], PD [58,135], and AD [136,137] as well as certain neuropsychiatric disorders [138]. In ALS, the pathogenic proteins TDP-43, FUS, and Chromosome 9 open reading frame 72 (C9Orf72) have been shown to have direct roles in DNA repair, in addition to their well-known contribution to its pathophysiology [139]. Respectively, TDP-43 and FUS are DNA- and DNA/RNA-binding proteins that are mutated in approximately 5–10% of ALS patients [69,71], and both proteins interact with the transcription-coupled nucleotide excision repair machinery [72]. Most ALS cases (97%) exhibit TDP-43 proteinopathy that is characterized by mislocalization and aggregation of the protein in the neuronal cytoplasm. A loss of nuclear TDP-43 in the mouse brain leads to impaired DNA repair, increased DSBs, inflammation, and neuronal senescence [71], features frequently observed in ALS brains [140]. FUS forms a complex with PARP1, XRCC1, and DNA ligase 3 to repair nuclear oxidative DNA damage and SSBs [141] and recruits mtDNA ligase 3 to resolve mtDNA damage [76]. Thus, FUS plays an important role in maintaining both nuclear and mtDNA integrity. A large study of motor neuron and spinal cord samples from ALS patients identified upregulation of p53 in both tissues, with upregulation being the greatest for patients with *C9Orf72*-repeat expansions [52]. Thus, genomic instability (i.e., splicing alterations, somatic mutations, gene fusions) appears to contribute to persistent DDR and p53 signaling in ALS, especially in those with TDP-43 proteinopathy.

There is also substantial evidence that α-synuclein, the pathological hallmark of PD, but also found in ALS [142] and AD [143], regulates the repair of DNA damage both in neural and cancer tissues [144,145]. Using a combination of electrophoretic mobility shift assays and atomic force microscopy, Dent and colleagues [146] showed that α-synuclein (vs. *β*- and γ-synuclein) binds in vitro to DNA and bends it into a more stable form, but not when it is phosphorylated. Phosphorylated α-synuclein is a toxic form, leading to its oligomerization and subsequent neurodegeneration in the PD brain [147], potentially in part because modified synuclein fails to bind to DNA. The oxidized form of α-synuclein also induces DNA strand breaks, while wild-type α-synuclein appears to regulate DDR signaling [135,148] as well as the repair of DSBs [144] and oxidative DNA damage [61,149] in both animal models of PD and in postmortem brain tissue. Like tau, nuclear α-synuclein is also important for repairing DSBs in cancer cells [145]. Thus, α-synuclein has both neuroprotective (DNA binding) and neurotoxic properties (phosphorylated and oxidized forms), with the relative importance of these paradoxical roles (regulation of DNA damage and toxic modified forms) beginning to become clearer [148].

One of the characteristic pathological hallmarks of AD brains is the presence of intracellular neurofibrillary tangles composed of hyperphosphorylated tau [150,151]. Like unmodified α-synuclein, recent studies indicate that unmodified nuclear tau has a neuroprotective role in the healthy brain [49,152,153,154,155]. Nuclear tau appears to be neuroprotective by binding DNA and preventing damage induction and by potentiating the DDR response in neurons [153,154,156,157]. Tau also appears to affect the response of cancer cells to DNA-damaging agents [158,159], possibly by regulating the nuclear trafficking of DNA repair proteins. Tau was recently shown by Asada-Utsugi and colleagues [153] to colocalize with DSBs in the AD brain, and when tau was ‘knocked down’ in mouse neurons, there was increased accumulation of DSBs, implicating tau in the regulation of DNA repair efficiency. Thus, nuclear tau, like pathological proteins in other neurodegenerative diseases (i.e., FUS, TDP-43, α-synuclein), appears to play an important role in maintaining the genomic stability of both post-mitotic neurons and cancer cells [151,159].

### 1.9. Genomic Instability in Neuropsychiatric Disorders

Development of the human brain requires the coordination of the proliferation, differentiation, and migration of neural stem cells and neuroprogenitors before their regional incorporation in the CNS [160]. Among embryonic tissues, the developing brain appears to be the most sensitive to DNA damage, especially during the expansion of the neuroprogenitor pool at the early stages of brain development [161]. DDR and the DNA repair machinery play pivotal roles in maintaining genome integrity during brain development by detecting and resolving DNA damage of both endogenous and exogenous origin [160,162,163]. Failure of the DDR or DNA repair machinery during brain development often leads to neurodevelopmental disorders [164], and failure during key stages of neurodevelopment could lead to de novo mutations, which predominantly occur in certain neuropsychiatric disorders (e.g., ASD). The accumulation of endogenous or environmentally-induced DNA damage, especially at different stages of neurogenesis, can impair proper brain development leading to long-term neurodevelopmental deficits [85,162,165,166,167].

While data on the accumulation of DNA damage in the nuclear and mitochondrial genomes of individuals affected by neuropsychiatric disorders are reasonably robust, studies connecting the increased genomic damage to impaired DNA repair capacity are limited or incomplete. While available data imply that defects in DNA repair contribute to disease etiology, direct evidence for impaired DNA processing in these neuropsychiatric disorders is lacking. Investigations designed specifically to assess DNA repair capacity in peripheral lymphocytes using a variety of methods have failed to reveal differences between subjects with and without schizophrenia [168,169], although potential defects in DDR signaling have been described [170]. Moreover, keeping in mind the small sample size, a more recent investigation using the Comet assay has suggested that drug-naïve patients with schizophrenia, positive family history, and longer duration of illness do, in fact, exhibit significantly decreased capacity to repair DNA damage in circulating peripheral lymphocytes [171].

Neuropsychiatric disorders, such as schizophrenia, autism spectrum disorder (ASD), and bipolar depression, are frequent, multi-factorial, and multi-symptomatic disorders [172]. A common feature observed among these neuropsychiatric diseases is disturbance of oxidative metabolism (i.e., oxidative stress) and maintenance of genomic integrity (e.g., DNA repair processes) during critical periods of brain development [173,174]. Ample evidence implicates oxidative stress, deficient repair of oxidative DNA lesions, and DNA damage in the development of these disorders [166,175,176,177]. A large meta-analysis of de novo mutations in the human genome due to errors in DNA repair or replication shows their association with developmental and psychiatric disorders [178]. As tools come on board to assess DNA repair capacity in clinically relevant biological samples, future studies have the potential to establish the likely contribution of DNA repair defects and genomic stress in neuropsychiatric disorders. We highlight the current evidence implicating impaired DNA repair and genomic stress in three prominent psychiatric diseases.

***Schizophrenia:*** Increased repair of DSBs by HRR and oxidative DNA damage appears to be a distinguishing feature in the brains of patients with schizophrenia who were exposed to environmental stressors (psychosocial stress) [138,172]. Neural stem cells (NSCs) and neurons derived from fibroblasts of patients with schizophrenia revealed differentially expressed proteins in DNA repair (NHEJ, MMR) and nuclear DDR proteins, respectively. Those patients with a family history of schizophrenia were enriched for genes regulating the HHR pathway (*MRE11*, *BRCA2*, *ATRX*, *RPA1*, *POLA1*, *LIG1*, *GEN1*) and accumulated DSBs in brain cells. Moreover, genotyping studies have found that certain single-nucleotide polymorphisms in genes that encode proteins involved in BER, HHR, or NER, i.e., *OGG1*, *XRCC1*, *XRCC3*, and *XPD*, are associated with schizophrenia [179]. Collectively, these data imply that genomic stress appears to be a characteristic feature in the brain of the MAM animal of schizophrenia and certain patients with schizophrenia.

***ASD.*** The molecular pathogenesis of ASD is complex, but recent human and mouse models point to an underlying role of DNA damage and perturbed DNA repair, as well as altered epigenetic mechanisms, in disrupting normal brain development [175,180,181,182]. Recent advances in the mapping of DNA damage in the human genome have provided insight into the vulnerable genes and associated pathways in both neurodegenerative diseases (e.g., AD) and neuropsychiatric disorders, notably ASD [183]. In particular, basal levels of oxidative DNA strand breaks are greater in autistic than non-autistic children [184]. There are also significant differences in the DNA repair capacity of peripheral blood mononuclear cells from parents of children with ASD compared to a control group [180]. Whether differences in DNA repair capacity occur in the brains of both the parents and their ASD children requires further investigation. Additionally, there is emerging evidence that ATM, which orchestrates the repair of DSBs, plays an important role in the development of the GABAergic system, the main inhibitory neurotransmitter circuitry in the brain [38,185].

***Bipolar Disorder.*** Patients with bipolar depression exhibit persistent oxidative stress-induced DNA damage that appears to have a major role in the pathophysiology of this neuropsychiatric disorder [166,177,186,187,188]. Oxidative stress-induced DNA and RNA damage (8-oxodG) was ~22% and 14% higher, respectively, in the urine of patients with newly diagnosed bipolar disorder compared to their unaffected relatives [188]. In euthymic bipolar patients (i.e., neither manic/hypomanic nor depressed), along with elevated levels of specific oxidative base lesions, *OGG1* transcript levels were reported to be significantly lower than those in healthy individuals, with the two groups exhibiting similar levels of *NEIL1* expression [189]. A recent comprehensive literature search has revealed associations of BER SNPs with bipolar disorder and major depressive disorder, although the data are limited and often conflicting or incomplete [177]. Czarny and colleagues have reported that DNA repair of oxidative DNA damage in peripheral blood mononuclear cells of subjects with depressive disorder is reduced in comparison to samples from control subjects [190,191]. To assess the role of DNA repair defects and DNA damage accumulation in the development of neuropsychiatric phenotypes, Mueller and colleagues employed a mouse model where the core BER factor, XRCC1, was conditionally knocked-out in the forebrain of postnatal animals [175]. While motor coordination, cognition, and social behavior remained unchanged, *XRCC1* inactivation in the dorsal dentate gyrus, CA1 and CA2, and the amygdala, caused increased DNA damage and anxiety-like behavior in males, but not in females.

### 1.10. Role of Environmental Factors in Neuropsychiatric Disorders

Environmental factors that disrupt early-life genomic stability in the brain (e.g., air pollution, carcinogens) may be key triggers of neuropsychiatric disorders [85,192,193,194,195]. For example, several mutagens, including radiation and polycyclic aromatic hydrocarbons, disproportionately were found to mutate long genes (in induced pluripotent stem cells) related to neurodevelopmental disorders, e.g., ASD, schizophrenia, and attention-deficit hyperactivity disorder. Genes involved in neuron projection guidance, sensory organ morphogenesis, and neuronal differentiation were the most vulnerable to various environmental mutagens. The effect seemed to be somewhat specific in that other disease-related genes, including those associated with ALS and AD, were not mutated by the mutagens more than expected [85]. ASD is modeled in the *BTBR* (Black and Tan Brachyury) inbred mouse strain mouse (BTBR T^+^ Itpr3tf/J), which exhibits social deficits characterized by poor social interaction and impaired communication, as well as repetitive stereotype behaviors and atypical vocalization [196]. Treatment of BTBR animals with the mycotoxin aflatoxin B1 (a potent carcinogen and food contaminant) resulted in increased micronuclei generation, oxidative DNA strand breaks, and apoptosis [167], as well as behavioral and immunological abnormalities [197]. AFB_1_ exposure appeared to intensify the neurobehavioral and immunological abnormalities in BTBR mice and also downregulate the expression of oxidative DNA repair genes (i.e., *OGG1*, *XRCC1*) in BTBR mice [167]. Additional risk factors that affect the development of ASD include advanced parental age at the time of conception, maternal obesity, diabetes or immune system disorders, prenatal exposure to air pollution or certain pesticides, extreme prematurity or very low birth weight, and oxygen deprivation at birth.

As yet another example of a genotoxin exposure being connected to neuropsychiatric symptoms, mice treated with MAM, which is etiologically linked with Western Pacific ALS/PDC (*vide supra*), produces a highly reproducible and reliable animal model of schizophrenia [198,199,200]. The MAM animal model of schizophrenia replicates both changes in mesolimbic dopamine function, which may contribute to the positive symptoms of schizophrenia, and altered frontal cortical–limbic circuits thought to be associated with the negative and cognitive impairments of the human disorder [34]. Schizophrenia-like deficits develop in the juvenile offspring of pregnant mice or rats treated with a carefully timed in utero dose (gestational day 16 or 17) of MAM [199,201]. Thus, exposure of the developing rodent brain to a single dose of MAM (neurodevelopmental model) recapitulates the histological, neurophysiological, and behavioral deficits observed in human schizophrenia [199,202]. A single administration of MAM also produces both oxidative- and alkylation-induced DNA damage [33,203] and epigenetic changes in the developing rodent brain [34]. Since repair sites are predominantly located in neuronal enhancers at CpG DNA methylation [204,205], the elevated DNA damage and epigenetic changes that occur in the MAM animal model of schizophrenia may be linked.

## 2. Conclusions and Future Directions

While several genetic and environmental factors have been implicated in the pathogenesis of neurodegenerative and neuropsychiatric diseases, emerging evidence indicates that genomic stress, caused by elevated endogenous or exogenous chemicals with genotoxic potential and/or defective DNA repair, may be an early event that promotes the accumulation of pathological proteins in the brain, evolving dysfunction and loss of neurons, and disruption of neuronal networks, leading to progressive loss of brain function [137,206,207]. Though substantial evidence has accumulated to indicate that loss of genomic integrity is a cardinal event in many human neurodegenerative diseases (e.g., ALS, PD, AD, and related disorders), more research is needed to determine the relative importance of genomic stress in triggering the underlying progressive process of neurodegeneration and whether the most critical pathogenic events occur early (e.g., during neurogenesis) or later in life (i.e., when cells are largely terminally differentiated). Equally important is whether the loss of genomic integrity during critical periods of brain development plays an important role in the etiopathogenesis of certain neuropsychiatric disorders, such as schizophrenia, ASD, and bipolar disorder. Although emerging evidence suggests an important role for DNA repair during the early stages of human brain development (e.g., in utero, postnatal), many questions remain. Are the cellular processes that repair DNA damage during development similar across brain regions and cell types (e.g., neuroprogenitors vs. glial progenitors)? Are there differences observed in DNA damage/repair during different stages of brain development? From a life course perspective, are early-life complications more critical to the development of brain dysfunction than harmful or sporadic events that take place later in life, such as during adulthood? What is the role of endogenous and environmental chemicals with the potential to induce genomic stress? As tools come on board to assess genome-wide DNA damage profiles and DNA repair capacity in more clinically relevant biological samples, future studies have the potential to more conclusively establish the contribution of genomic stress and DNA repair defects in diseases of the nervous system.

## Figures and Tables

**Figure 1 ijms-25-07221-f001:**
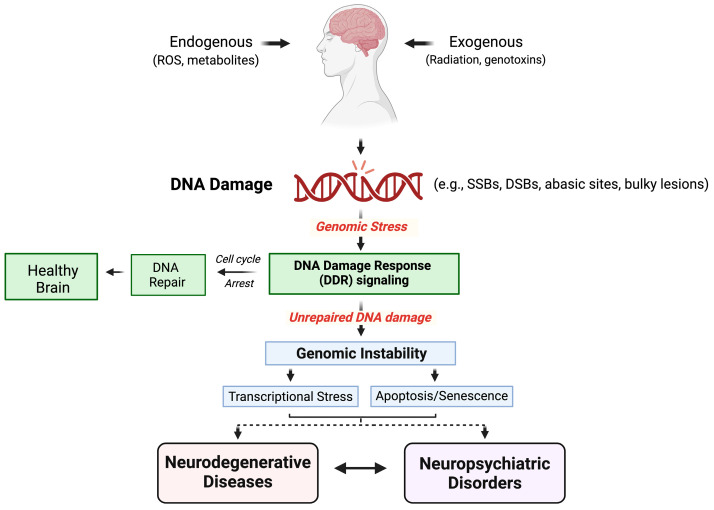
Factors that contribute to genomic stress in both neurodegenerative diseases and neuropsychiatric disorders. **ROS**, reactive oxygen species; **SSB**, single-strand breaks; **DSB**, double-strand breaks.

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
