# Peer review of "Introducing the Role of Genotoxicity in Neurodegenerative Diseases and Neuropsychiatric Disorders"

_ijms, 2024, doi:10.3390/ijms25137221_

Round 1

Reviewer 1 Report

Comments and Suggestions for Authors

The authors presented for consideration a review devoted to the role of genotoxicity in the development and emergence of neurodegenerative and neuropsychiatric diseases. The factors influencing the stability of DNA (and factors that can influence DNA, for example, aging, environmental situation, etc.) and, as a consequence, the occurrence of genomic abnormalities are considered. The review may be useful for choosing possible strategies when developing interventions aimed at preventing brain diseases. The authors examined neurodegenerative and neuropsychiatric disorders both separately and together, which is of interest for further research into the causes of these diseases. The references are mainly represented by articles published within the last 5 years. At the moment, the review contains modern and relevant information, both on genomic diseases and on the causes of their occurrence (for example, DNA replication disorders, etc.). However, I have a few comments that could improve this review.

1. The authors paid little attention to the issue of double and single-strand DNA breaks, as well as their length, on the influence of the incidence of genetic diseases.

2. It is advisable to add a diagram describing “cross” diseases of different groups (neurodegenerative and neuropsychiatric), i.e. how often these groups of diseases manifest themselves in one person or his relatives.

3. The purpose of this review should be added to the introduction.

In general, the review is written quite interestingly and in detail. May be published after minor corrections.

Author Response

The authors presented for consideration a review devoted to the role of genotoxicity in the development and emergence of neurodegenerative and neuropsychiatric diseases. The factors influencing the stability of DNA (and factors that can influence DNA, for example, aging, environmental situation, etc.) and, as a consequence, the occurrence of genomic abnormalities are considered. The review may be useful for choosing possible strategies when developing interventions aimed at preventing brain diseases. The authors examined neurodegenerative and neuropsychiatric disorders both separately and together, which is of interest for further research into the causes of these diseases. The references are mainly represented by articles published within the last 5 years. At the moment, the review contains modern and relevant information, both on genomic diseases and on the causes of their occurrence (for example, DNA replication disorders, etc.). However, I have a few comments that could improve this review.

  1. The authors paid little attention to the issue of double and single-strand DNA breaks, as well as their length, on the influence of the incidence of genetic diseases.

Response: We thank the reviewer for comments to improve the overview of this understudied area of research.  We have included more detailed information about both DSBs and SSBs in the section on Genomic Instability in Neurodegenerative disease (Lines 151-157).  

  1. It is advisable to add a diagram describing “cross” diseases of different groups (neurodegenerative and neuropsychiatric), i.e. how often these groups of diseases manifest themselves in one person or his relatives.

Response: The focus of this Special Issue is on genotoxicity in both neurodegenerative diseases and neuropsychiatric disorders and not about the disorders themselves.  Thus, respectfully, the authors agree that the addition of a diagram is beyond the scope of this current review of the topic.  However, in seeking to respond to the reviewer’s concern, we have added a paragraph describing additional overlapping features in multiple diseases in addition to the existing focus of ALS and schizophrenia.

  1. The purpose of this review should be added to the introduction.

Response: We have made the necessary changes in the Title (Line 1), Abstract (Lines 15, 21), and the Introduction (Lines 59-61) to reflect the purpose of this review. 

In general, the review is written quite interestingly and in detail. May be published after minor corrections.

Reviewer 2 Report

Comments and Suggestions for Authors

The article titled "Role of Genotoxicity in Neurodegenerative Diseases and Neuropsychiatric Disorders," examines the role of genotoxicity (DNA damage) in both neurodegenerative diseases (ALS, PD, AD) and neuropsychiatric disorders (schizophrenia, autism, bipolar disorder) with major focus on DNA damage response (DDR) and its dysregulation, discussing both endogenous and exogenous factors contributing to DNA damage.

The topic is highly relevant to current research in molecular neuroscience and neurogenetics, addressing a significant intersection between neurodegeneration and psychiatric conditions, and offers insights into the role of DNA damage in these pathologies.

Major concerns:

·      In some sections the manuscript lacks specificity, making unclear the goal of the review

·      While the paper provides a broad overview, some sections lack detailed mechanistic insights.

·      The discussion on neuropsychiatric disorders could be more robust, including more detailed analysis and recent studies

·      The discussion on genomic instability in neurodegenerative disorders should be addressed and discussed in deeper details

Author Response

The article titled "Role of Genotoxicity in Neurodegenerative Diseases and Neuropsychiatric Disorders," examines the role of genotoxicity (DNA damage) in both neurodegenerative diseases (ALS, PD, AD) and neuropsychiatric disorders (schizophrenia, autism, bipolar disorder) with major focus on DNA damage response (DDR) and its dysregulation, discussing both endogenous and exogenous factors contributing to DNA damage.

The topic is highly relevant to current research in molecular neuroscience and neurogenetics, addressing a significant intersection between neurodegeneration and psychiatric conditions, and offers insights into the role of DNA damage in these pathologies.

Major concerns:

  • In some sections the manuscript lacks specificity, making unclear the goal of the review.

Response: We have made the necessary changes in the Title (Line 1), Abstract (Lines 15, 21), and the Introduction (Lines 59-61) to reflect the purpose or goal of this review.  

  • While the paper provides a broad overview, some sections lack detailed mechanistic insights.

Response: The main focus of this paper is to highlight recent work on genomic integrity in both neurodegenerative disease and neuropsychiatric disorders.  We included more detailed information on the overlap of neurodegenerative and neuropsychiatric disorders (Lines 67-77) as well as a more detailed discussion of DSBs and SSBs in inherited DNA-repair syndromes (Lines 151-157). 

  • The discussion on neuropsychiatric disorders could be more robust, including more detailed analysis and recent studies.

Response: As we state above, the main focus of this review is to highlight recent work on genomic stress in neuropsychiatric disorders, not to provide a comprehensive review.  As we pointed out in the section on Genomic Instability in Neuropsychiatric disorders (Lines 377-378 and 392-395), the data are limited or incomplete and more research needs to conducted to demonstrate that genomic stress is an important trigger of psychiatric disorders (Lines 476-488). The need for further research targeted on this subject is now added to the Conclusion.

  • The discussion on genomic instability in neurodegenerative disorders should be addressed and discussed in deeper details.

Response: We have included more detailed information about both DSBs and SSBs in the section on Genomic Instability in Neurodegenerative disease (Lines 151-157).  

Reviewer 3 Report

Comments and Suggestions for Authors

The manuscript of the review article entitled “Role of Genotoxicity in Neurodegenerative Diseases and Neuropsychiatric Disorders” represents an excellent summary of the potential role of genotoxicity during the development of these diseases.

The authors revise very recent papers and guide the reader in understanding the potential importance of genotoxicity, which has not been established so far as a major cause involved in or triggering the development of these diseases.

However, some of the papers cited describe quite preliminary findings since a very small number of subjects is included in the studies, for example, in the case of DSB in AD, where only three patients and 3 controls were analyzed.

Also, in the study that reported a higher accumulation of 8-oxodG and S42Y alpha-synuclein, 8 patients and 9 control brains were included, and mutated alpha-synuclein mRNAs represented 6.7% of total alpha-synuclein mRNAs in controls and 25.4% in PD.

These data underline the importance of genotoxicity; however, I would suggest discussing its role in developing neurodegenerative diseases more carefully. It should be presented as an understudied mechanism, which could be one of the several risk factors leading to the development of neurodegeneration. In this sense, the sentence between lines 450-453 should be rewritten, maybe only eliminating “overwhelming” and substituting “is a cardinal event” with “occurs in”.

Author Response

The manuscript of the review article entitled “Role of Genotoxicity in Neurodegenerative Diseases and Neuropsychiatric Disorders” represents an excellent summary of the potential role of genotoxicity during the development of these diseases. 

The authors revise very recent papers and guide the reader in understanding the potential importance of genotoxicity, which has not been established so far as a major cause involved in or triggering the development of these diseases.

However, some of the papers cited describe quite preliminary findings since a very small number of subjects is included in the studies, for example, in the case of DSB in AD, where only three patients and 3 controls were analyzed. 

Response: We thank the reviewer for comments to improve the overview of this understudied area of research.  In response, we have included the number of subjects studied by Zhang et al (2023) and provided additional references on the role of DSBs in AD (Line 177).  

Also, in the study that reported a higher accumulation of 8-oxodG and S42Y alpha-synuclein, 8 patients and 9 control brains were included, and mutated alpha-synuclein mRNAs represented 6.7% of total alpha-synuclein mRNAs in controls and 25.4% in PD.

Response: We thank the reviewer for pointing this out.  We have revised this section (Lines 189-194) to reflect these preliminary findings.  

These data underline the importance of genotoxicity; however, I would suggest discussing its role in developing neurodegenerative diseases more carefully. It should be presented as an understudied mechanism, which could be one of the several risk factors leading to the development of neurodegeneration. In this sense, the sentence between lines 450-453 should be rewritten, maybe only eliminating “overwhelming” and substituting “is a cardinal event” with “occurs in”.

Response: We thank the reviewer for comments to improve the discussion on the development of neurodegenerative disease.   We removed overwhelming and replaced it with substantial (Line 470).  We have indicated that more research needs to be done to determine the relative importance of genomic stress in the pathological process of neurodegeneration, especially during early or later stages of life (Lines 472-474).